# Increased Frequency of Circulating Activated FOXP3^+^ Regulatory T Cell Subset in Patients with Chronic Lymphocytic Leukemia Is Associated with the Estimate of the Size of the Tumor Mass, STAT5 Signaling and Disease Course during Follow-Up of Patients on Therapy

**DOI:** 10.3390/cancers16183228

**Published:** 2024-09-22

**Authors:** Zlatko Roškar, Mojca Dreisinger, Evgenija Homšak, Tadej Avčin, Sebastjan Bevc, Aleš Goropevšek

**Affiliations:** 1Department of Haematology, University Medical Centre Maribor, 2000 Maribor, Slovenia; zlatko.roskar@ukc-mb.si (Z.R.); mojca.dreisinger@ukc-mb.si (M.D.); 2Department of Laboratory Diagnostics, University Medical Centre Maribor, 2000 Maribor, Slovenia; evgenija.homsek@ukc-mb.si (E.H.); ales.goropevsek@ukc-mb.si (A.G.); 3Faculty of Medicine, University of Maribor, 2000 Maribor, Slovenia; 4Department of Allergology, Rheumatology and Clinical Immunology, Children’s Hospital, University Medical Centre Ljubljana, 1000 Ljubljana, Slovenia; tadej.avcin@kclj.si; 5Department of Pediatrics, Faculty of Medicine, University of Ljubljana, 1000 Ljubljana, Slovenia; 6Department of Nephrology, University Medical Centre Maribor, 2000 Maribor, Slovenia

**Keywords:** chronic lymphocytic leukemia (CLL), signal transduction, cytokines, T cells

## Abstract

**Simple Summary:**

Patients with chronic lymphocytic leukemia (CLL) are more susceptible to infections, which are also the most common cause of death in these patients. Previous studies in patients with CLL described elevated levels of FOXP3^+^ regulatory T cells (Tregs), which also correlated with decreased T cell responses to microbial antigens. As the activation of the STAT5 transcription factor induces the expression of FOXP3 and human CD4^+^FOXP3^+^ T cells that also contain nonsuppressive T cells, we analyzed STAT5 phosphorylation (pSTAT5) and suppressive subpopulations, including activated Tregs (aTregs). We found a significantly increased frequency of aTregs in patients with advanced stages, which significantly correlated with the total tumor mass score. aTreg expansion in vitro was associated with significantly higher aTreg pSTAT5 responses to SARS-CoV-2 antigen-specific stimulation in vitro. Finally, a subgroup of patients characterised by an increased aTreg percentage among CD4^+^FOXP3^+^ T cells experienced a more severe disease course with serious grade ≥3 infections during follow-up.

**Abstract:**

Introduction: Advanced chronic lymphocytic leukemia (CLL) is accompanied by increased circulating regulatory T cells (Tregs) and increased susceptibility to severe infections, which were also shown to entail a striking induction of FOXP3 expression in Tregs. As homeostasis of the most suppressive CD45RA^−^FOXP3^high^ activated Treg (aTreg) subset differs, it is critical to analyse homeostatic signalling in Treg subsets. Materials and Methods: In this study, by using conventional and imaging flow cytometry, we monitored STAT5 signalling/phosphorylation (pSTAT5) and investigated Treg subsets in relation to the Binet stage, the total tumor mass score (TTM) and the disease course during a follow-up of 37 patients with CLL. Results: The aTreg percentage was significantly increased among CD4^+^ T cells from patients with advanced disease and significantly correlated with the TTM. A subgroup of patients with higher aTreg percentages among CD4^+^FOXP3^+^ T cells at the start of therapy was characterised by more frequent episodes of severe infections during follow-up. Conclusions: The results suggesting that an aTreg fraction could represent a possible marker of a severe disease course with infectious complications. Augmented homeostatic STAT5 signalling could support aTreg expansion, as higher pSTAT5 levels were significantly correlated with an increased aTreg frequency among CD4^+^FOXP3^+^ T cells during the follow-up of patients on therapy, as well as following SARS-CoV-2 antigen-specific stimulation in vitro.

## 1. Introduction

Chronic lymphocytic leukemia (CLL), characterised by the accumulation of monoclonal B lymphocytes in bone marrow, lymphoid organs, and peripheral blood, is the most common form of leukemia in the Western countries [1]. CLL cells are microenvironment-dependent, and CLL progression is associated with the alteration of various immune cell populations to create a niche suitable for the proliferation and survival of leukemic B cells [2]. Patients with CLL have a higher risk of both more severe coronavirus disease (COVID-19) and higher mortality [3,4]. They are more susceptible to infections, which are also the most common cause of death in these patients [5]. The immunosuppressive microenvironment of CLL, which supports disease progression and contributes to CLL escaping immune surveillance, is also thought to significantly destabilise the whole immune response of the CLL patient [6]. Impairments in the regulation of the immune system that occur in CLL patients at an early stage and are exacerbated during disease progression, are also responsible for the poor response of these patients to infections with various pathogens [5,7,8]. Clonal change in B lymphocytes in CLL has a strong effect not only on susceptibility to infections, but also on the humoral antibody response to vaccines [9]. An adequate humoral immune response requires helper T cells (CD4^+^ T lymphocytes), which play an essential role in recruiting and activating B lymphocytes, which are responsible for antibody production. CD4^+^ T lymphocyte responses to viral spike protein have been shown to correlate with anti-SARS-CoV-2 IgG and IgA antibody titres [10].

A special subpopulation of CD4^+^ T lymphocytes are regulatory T lymphocytes (Treg), which express the characteristic transcription factor FOXP3 and show in vitro suppression [11,12]. Tregs represent an essential mechanism of peripheral T cell tolerance, which was first demonstrated in Treg-depleted animal models that develop diffuse autoimmunity [13].

Tregs were found to also play a major role in the development of an immunosuppressive tumor microenvironment in patients who had various cancers before [14] and to comprise functionally distinct subpopulations [15]. As shown in a recent study, signalling responsiveness to cytokines only in a specific-activated (aTreg) subset in peripheral blood, reflects intratumoral immunosuppressive potential and predicts future relapses [16].

The results of several studies indicate an increased proportion of total Treg cells in CLL patients compared to healthy controls and association with disease course [17,18,19,20]. However, the importance of different Treg subpopulations and their cytokine signalling in disease progression or response to treatment has not been definitively determined to date. In addition, in CLL patients, signalling responses to cytokines were studied mostly in clonal B cells, but not in CD4^+^ T cell subsets [21,22]. Homeostatic cytokines such as IL-2 and IL-7 act on Treg and other FOXP3-negative/conventional CD4^+^ cells through their intracellular signalling pathways originating from surface receptors, mainly through STAT5 (signal transducer and activator of transcription 5) proteins T (Tcon). When STAT proteins are activated, they are phosphorylated on specific tyrosine residues and translocate to the nucleus, where they control several gene programs that also regulate their proliferation [23]. Recent studies have demonstrated that Tregs are subject to more distinct homeostatic STAT5 signalling controls than Tcon subsets [24,25]. However, little is known of the nature of STAT5 signalling dysfunctions in CD4^+^ T cell subsets and their possible role in perturbed Treg/Tcon homeostasis, disease progression and susceptibility to infections in patients with CLL. 

The aim of our study was to investigate if the Treg subsets and STAT5 phosphorylation (pSTAT5) are altered in CD4^+^ T cells in the blood of patients with CLL and whether the changes in circulating Treg subsets are related to STAT5 signalling imbalances, to the Binet stage, to the estimate of the size of tumour mass and to the disease course during the long-term follow-up. By using phospho-specific flow cytometry, our analysis was designed to also monitor pSTAT5 during in vitro SARS-CoV-2 antigen-specific stimulation and to directly compare STAT5 activation in Tcon and activated Treg subsets in each patient.

## 2. Materials and Methods

### 2.1. Study Population

Fifty-six untreated consecutive patients meeting the diagnostic criteria for CLL [26] were enrolled into the single-centre, prospective cohort study.

Thirty-seven out of 56 patients who started therapy at enrolment (disease activity according to the international workshop on CLL-iwCLL criteria [26]) were followed up to 500 days. The demographic, clinical and laboratory data at the time of the follow-up study entry are presented in Table 1. The therapy during follow-up is described in Table 2. During the follow-up, an episode of grade ≥3 infection was defined as in the Common Terminology Criteria for Adverse Events (CTCAE) Version 5.0.

The malignant disease burden in patients with CLL was assessed with the Binet staging [26,27,28]. 

Blood samples were also taken from twenty healthy controls with a mean age of 60.3 years (minimum 51.4 and maximum 82.7 years) who had no history of allergies, acute infections, autoimmune disorders or immunosuppressive medicines.

The SARS-CoV-2 Ag-specific pSTAT5 response was, in addition to two patients included in the follow-up study, examined in 10 CLL patients (independent of their stage of the disease), who all received two doses of the BNT162b2 mRNA COVID-19 vaccination 26 to 28 months ago. Within a few months, a couple of them had also recovered from a recent SARS-CoV-2 infection.

Appendix A displays pertinent clinical and demographic information for the recruited patients that were not included in the follow-up study.

Blood samples were also taken from 12 healthy laboratory staff members for the SARS-CoV-2 antigen-specific pSTAT5 assay 26 to 32 months after the second dose of the BNT162b2 mRNA COVID-19 vaccine, or after they had recovered from SARS-CoV-2 infection.

### 2.2. Preparation of Whole-Blood Samples for Analysis of STAT5 Phosphorylation

EDTA anticoagulated whole blood was used to prepare samples for the flow cytometry investigation of basal and cytokine-induced STAT5 phosphorylation ex vivo. Generally, 100 µL of sample was either left untreated or treated with 100 ng/mL IL-2 (Peprotech, Rocky Hill, NJ, USA) for 15 min at 37 °C in a water bath. The maximal induction of pSTAT5 by T cells was observed earlier in IL-2 stimulation protocols at 15 min. The sample was placed in a round-bottom polystyrene test tube (Falcon^®^ 5 mL).

Whole-blood samples were fixed for 10 min using 2 mL of BD Phosflow Lyse/Fix Buffer (BD Biosciences, San Jose, CA, USA) to stop phosphorylation. After centrifuging the samples at 300× *g* for 7 min, the cells were permeabilised by letting them sit on ice in 1 mL of diluted BD Perm Buffer III (BD Biosciences) for half an hour.

### 2.3. Flow Cytometry Analysis after Staining with Antibodies Specific to T Cell Subsets and Phosphorylated STAT5 Tyrosine

Following permeabilisation, the samples were centrifuged for five minutes at 300× *g*, and then they were repeatedly washed with two millilitres of phosphate-buffered saline (PBS). Subsequently, the cells were stained for 30 min in 100 µL of stain buffer (PBS/2% FBS) using the following antibodies: anti-CD45-PerCP (5 μL, clone 2D1), BV786 (1 μL, clone HI30) or APC-Cy7 (5 μL, clone 2D1), anti-CD3-FITC (10 μL, clone UCHT1), BV650 (3 μL, clone SK7) or PerCP (20 μL, clone SK7), as well as antibodies that recognise particular phosphorylated STAT5 tyrosine: pSTAT5 (Y694)-Alexa647 (10 μL, clone 47) or PE (10 μL, clone 47) (all BD Biosciences). Antibodies specific to T cell subsets were used to simultaneously stain cells at room temperature for multiparametric immunophenotyping assays. These antibodies included anti-CD25-PE (10 μL, clone 2A3), BV421 (3 μL, clone 2A3) or APC (5 μL, clone 2A3), anti-CD4-PECy7 (5 μL, clone SK3) or BV750 (2 μL, clone SK3), anti-FOXP3-Alexa 488 (10 μL, clone 259D/C7) or PE (10 μL, clone 259D/C7) (all BD Biosciences), anti-Ki67 PE (10 μL, clone B56) or BV650 (3 μL, clone B56) and CD45RA PE-Cy7 (0.5 μL, HI100) or APC (5 μL, clone HI100). Finally, cells were washed with 2 mL of stain buffer and acquisition was performed on an LSR II or FACSymphony A3 Flow Cytometer (Beckton Dickinson, Franklin Lakes, NJ, USA). A typical panel used and analysed on the latter flow cytometer together with gating hierarchy is shown on the Appendix A.

Subsequent analysis, including the measurement of the median fluorescence intensity (MFI) of the pSTAT5-specific signal, was performed using FACSDiva software version 9.0 (Becton Dickinson) and FlowJo version 10.8 (TreeStar, Ashland, OR, USA, now part of BD Biosciences). 

The “fluorescence-minus-one plus isotype” (FMO + I) control was used to set the gate for CD25^+^ and Ki-67^+^ cells. The same antibodies were used for staining as in the full stain, with the exception of the anti-CD25 and anti-Ki-67 antibodies, which were replaced with an isotype control antibody labelled with the same fluorochrome.

### 2.4. In Vitro Stimulation of Purified CD4^+^ T Cells and pSTAT5 Inhibition with Neutralising Anti-IL-2 Antibodies

Peripheral blood mononuclear cells (PBMCs) were separated from peripheral blood by density gradient centrifugation with Histopaque 1077 (Sigma-Aldrich, St. Louis, Mo, USA). Purified CD4^+^ T cell populations were prepared from PBMCs using negative selection by magnetic cell sorting (BDImag Human CD4 T Lymphocytes Enrichment Set–DM, BD Biosciences, San Jose, CA, USA). CD4 T cells were further separated into CD25^+^ and CD25^−^ populations (BD IMag™ An-ti-Human CD25 Magnetic Particles), anti-CD3/CD28 stimulated with plate-bound an-ti-CD3 (5 μg/mL) and soluble anti-CD28 (1 μg/mL) in 96-well plates and cultured in complete RPMI (RPMI 1640 plus L-glutamine, penicillin, streptomycin; Thermo Fisher Scientific, Waltham, MA, USA) with 10% foetal calf serum (Thermo Fisher Scientific) for 48 h. For the STAT5 signalling analysis of peripheral blood mononuclear cells (PBMCs) and isolated CD4^+^ T cells, the preparation of cells was the same as described above for whole blood, except for the fixation of cells; BD Cytofix Fixation Buffer (BDPharmingen, BD Biosciences, San Jose, CA, USA) was used.

In selected experiments, the samples were incubated with a neutralising antibody, an-ti-IL-2 (2 µg/mL, clone MQ1–17H12, BD Biosciences), for 30 min at 37 °C before fixation.

### 2.5. Flow Cytometric Analysis of pSTAT5 in Treg Subsets after Whole-Blood Stimulation with SARS-CoV2-Specific Antigens 

Using heparinised antigen tubes from the QuantiFERON SARS-CoV-2 kit (Qiagen, Hilden, Germany), we assessed STAT5 signalling responses to whole-blood antigen-specific stimulations. Epitopes from the S1 subunit of the CD4^+^ T cell spike protein are present in the SARS-CoV-2 Ag1 tube and epitopes from the S1 and S2 subunits of the CD4^+^ and CD8^+^ T cell spike protein are present in the SARS-CoV-2 Ag2 tube.

After being taken straight into the test collecting tubes, whole-blood samples were agitated and allowed to incubate for 16–24 h.

Before centrifugation and preparation for STAT5 signalling analysis, whole-blood aliquots (120 μL) were taken from Nil (negative control) and the two Ag tubes (60 μL each, mixed together) of the QuantiFERON SARS-CoV-2 kit. As described above, for the analysis of basal and cytokine-induced STAT5 phosphorylation ex vivo, fixing the samples with 2 mL of BD Phosflow Lyse/Fix Buffer (BD Biosciences, San Jose, CA, USA) for 10 min stopped the phosphorylation process. After that, samples were centrifuged at 300× *g* for 7 min and incubated for 30 min in 1 mL of ice-cold BD Perm Buffer III (BD Biosciences) to permeabilise the cells. Following permeabilisation, the samples were centrifuged for five minutes at 300× *g*, and repeatedly washed with two millilitres of PBS. The cells were then stained with the following combination of anti-human fluorescent monoclonal antibodies (all from Becton Dickinson, San Jose, CA, USA) for 30 min at room temperature in 100 µL of stain buffer (PBS/2% FBS): anti-pSTAT5-Alexa647, FOXP3 FITC, CD4 BV750, CD45RA PE-Cy7 (3 μL, HI100), CD25 BV421 and CD3 BV786 (2 μL, clone UCHT1). The FACSymhony A3 flow cytometer was used to analyse the cells, and FlowJo software version 10.8 (TreeStar) and FacsDiva software version 9.0 (Becton Dickinson, San Jose, CA, USA) were utilised for data analysis.

### 2.6. Imaging Flow Cytometry Analysis 

After preparing and staining whole-blood samples in the manner previously mentioned for CD3, FOXP3, CD4 and pSTAT5, 20 ng/mL 7-AAD (BD Biosciences) was used as a counterstain. Using the ImageStreamX imaging flow cytometer (Amnis, Seattle, WA, USA), image files of each sample were acquired and IDEAS software (Amnis) was used for analysis, as previously described [29]. Gating on 7-AAD-positive events with high nuclear aspect ratios (minor to major axis ratio, a measure of circularity) and high nuclear contrast (as determined by the Gradient Max feature) allowed for the identification of in-focus single cells. FOXP3^+^ cells were gated among lymphocytes (low side scatter/low area cells). The Similarity score, which quantifies the/correlation of pixel values of the nuclear and pSTAT5 images on a per-cell basis, allowed for the assessment of the nuclear localisation of pSTAT5 within these cells [30]. The two images will be similar and have large positive values of the Similarity score if the pSTAT5 is nuclear. In contrast, the images of the nucleus and pSTAT5 will be anti-similar and have negative-value Similarity scores if pSTAT5 is cytoplasmic. If events with positive Similarity score values > 1 were verified to calculate the proportion of the cells characterised by nuclear-localised pSTAT5 in the FOXP3-expressing population, they also showed visually evident nuclear distributions of the transcription factor.

### 2.7. Statistical Analysis

GraphPad Prism software version 10 for Windows (San Diego, CA, USA) was used to conduct a statistical analysis. The Wilcoxon matched pairs signed rank test was used for within-group comparisons, while the Mann–Whitney test was employed for between-group comparisons. The Spearman correlation coefficient was computed to investigate potential correlations between variables. *p* values were deemed significant if they were less than 0.05. The Kaplan–Meier survival method was used to analyse disease course/infectious complications in the follow-up period, and a Log-rank test was used to determine the significance between groups. When comparing outcomes and related clinical factors between subgroups, the Fisher’s exact test or the non-parametric Mann–Whitney test was used for analysis. Where necessary, the Bonferroni correction for multiple testing was implemented to account for multiple testing.

## 3. Results

### 3.1. The Increase in Activated Treg Subsets in Peripheral Blood from CLL Patients with Untreated Advanced Disease Correlates with Total Tumour Mass (TTM) Scoring

First, FOXP3 expressing CD4^+^ T cells were analysed by flow cytometry in whole-blood samples from patients with CLL and healthy controls (HC). The percentage of FOXP3^+^ cells among CD4^+^ T cells was significantly higher in patients with CLL (Figure 1A).

Next, Treg analysis was also performed by using the strategy introduced by Miyara et al. [15], allowing the functional delineation of the FOXP3 high-expressing CD45RA^−^FOXP3^hi^-activated Treg (aTreg) subset and the two FOXP3 low-expressing subsets: the CD45RA^+^FOXP3^lo^ resting Treg (rTreg) and the CD45RA^−^FOXP3^lo^ (non-Treg) subset (Figure 1B).

The frequency of all FOXP3^+^ subsets among CD4^+^ T cells was significantly increased in untreated CLL patients compared to HC (Figure 1C). 

As Treg suppressive function is linked to high surface expression levels of IL-2Ralpha (CD25) [31], percentages of CD25 expressing cells were compared between the subsets of FOXP3^+^ cells from patients with CLL. When compared to rTreg and nonTreg subsets, significantly higher percentages of CD25^+^ cells were found among aTregs (Appendix A), which were shown before to be the most suppressive Treg subset among FOXP3^+^ cells [15,32]. 

According to their stage of illness, patients with CLL are categorised into one of three groups, A–C, by the Binet classification [26]. Those with untreated stage C disease did not exhibit elevated values of rTreg, a nonTreg subset, or all the FOXP3^+^ cells among the CD4^+^ T cells compared to the stage A patient population when we examined the proportion of all FOXP3^+^ cells and their subsets among CD4^+^ T cells in patients at different stages of disease. However, we found a significantly increased proportion of aTreg subsets (Figure 1D).

As these results showed an expansion of FOXP3^+^ cells in CLL patients, with a skewing to the aTreg subset in advanced disease, the frequencies of the populations analysed were also correlated with lymphocyte counts. The percentage of aTreg subsets, defined as either CD45RA^−^FOXP3^hi^ or CD45RA^−^CD25^hi^ CD127^lo/−^ cells (Appendix A) among CD4^+^ T cells was significantly positively correlated with the lymphocyte count (Figure 1E), suggesting that this specific subset is associated with a malignant B cell burden in patients with CLL.

In order to assess the tumour mass within all major body compartments, we used the total tumour mass scoring system (TTM) in our patients with CLL [28]. TTM is the sum of: (1) the square root of the number of peripheral blood lymphocytes per nL, (2) the diameter of the largest palpable lymph node in centimetres, and (3) the enlargement of the spleen below the left costal margin in centimetres.

The aTreg subset, again defined as either CD45RA^−^FOXP3^hi^ or CD45RA^−^CD25^hig^CD127^lo/−^ cells, was significantly positively correlated with the TTM score (Figure 1F). There was a significant positive correlation between the TTM score and percentage of this subset, but not in all FOXP3^+^ cells among CD4^+^ T cells.

### 3.2. Increased Proportions of aTregs among FOXP3^+^CD4^+^ T Cells Are Associated with Their Augmented STAT5 Signalling Responses Following Whole-Blood SARS-CoV-2 Antigen-Specific Stimulation

While work over the past 30 years has clearly documented an important role for homeostatic cytokine/IL-2-induced STAT5 signalling in shaping the development of Tregs, recent research indicates that IL-2-dependent STAT5 signalling is also critical for their suppressive function [24]. Since the translocation of STAT5 homodimers to the nucleus follows the cytokine-dependent activation/phosphorylation of STAT5 [23], we also employed imaging flow cytometry to investigate nuclear/cytoplasmic localisation of pSTAT5. After stimulation with recombinant human IL-2 for 15 min, predominant nuclear localisation of pSTAT5 was found in FOXP3-expressing cells (Figure 2A). The nuclear localisation of pSTAT5, measured using the Similarity score, was higher in both FOXP3 high- and FOXP3 low-expressing cells compared to FOXP3 negative cells, gated as shown in Appendix A; **** *p* < 0.0001.; *** *p* < 0.001; ** *p* < 0.01; * *p* < 0.05.

Compared to FOXP3 negative Tcons, whole-blood FOXP3^+^ Tregs were shown to display higher levels of IL-2-dependent signalling before [25]. As IL-2 is also produced during the antigen-specific activation of Tcons, the pSTAT5 levels in CD25^+^FOXP3^+^ Tregs and Tcons were analyzed following the SARS-CoV-2 antigen-specific stimulation of lymphocytes in our previous study [33]. In the current investigation, the combination of two antigen peptides specific to SARS-CoV-2 (SARS-CoV-2 Ag1 and Ag2) from the QuantiFERON SARS-CoV-2 kit was used to examine the pSTAT5 response of specific (rTreg and aTreg) Treg subsets. Such antigen-induced pSTAT5 levels were analysed in whole-blood samples from HC who had recovered from SARS-CoV-2 infection or had received the BNT162b2 mRNA COVID-19 vaccination. Furthermore, we employed this technique to examine the pSTAT5 levels in rTreg and aTreg subsets from twelve CLL patients who received two doses of the BNT162b2 mRNA COVID-19 vaccination; some of them had also recovered from a recent SARS-CoV-2 infection. 

Significantly higher pSTAT5 levels in a subset of aTregs (Figure 2B) and rTregs (Appendix A) were observed after stimulation with the SARS-CoV-2 specific spike peptide mix (+Ag) compared to negative control (Nil) samples from HC and CLL patients. However, when the pSTAT5 MFI fold change (pSTAT5 MFI in the stimulated tube divided by pSTAT5 MFI in the control tube) was used to measure STAT5 signalling responses [34], it was significantly higher in aTregs than in rTregs, even from both HC patients with CLL (Figure 2C). Furthermore, the percentages of aTregs (Figure 2D), but not rTreg (Appendix A), subsets among FOXP3^+^CD4^+^ T cells were significantly increased after stimulation with a mixture of SARS-CoV-2-specific spike peptide (+Ag) compared to negative control (Nil). Moreover, a significant correlation was found between aTreg STAT5 signalling responses (pSTAT5 MFI fold change) and the increase in fold change in the percentage of aTregs (Figure 2E).

These results indicate that aTreg-specific pSTAT5 responses, which are higher than in rTreg subsets, are also associated with the expansion of aTreg subsets among FOXP3^+^CD4^+^ T cells following whole-blood SARS-CoV-2 antigen-specific stimulation.

### 3.3. aTreg Subsets, Disease Course and STAT5 Signalling during Follow-Up

As the increase in Treg proportions with increased FOXP3 levels was also associated with poor outcomes in SARS-CoV-2 infections [35], we investigated whether CLL patients with higher aTreg percentages among FOXP3^+^CD4^+^ T cells at the start of follow-up would experience a more severe disease course with infectious complications requiring hospitalisation using the approach described before [36,37]. 

The CLL cohort, composed of 37 patients with mostly stage C disease before the start of treatment at enrolment and with subgroups defined based on their aTreg percentages, (Table 1) was followed up to 500 days. The cutoff value for the percentage of activated Tregs was determined as the means + SEM (26,3%) in HCs. We found that 58% CLL patients in group 2, with higher aTreg frequencies (≥26%) experienced grade ≥3 infections by 365 days after enrolment, compared to only 6% of patients in group 1 with lower aTreg frequencies (Figure 3A). The patients in the former group had more grade ≥3 infections per year of follow-up than patients with lower aTreg frequencies: grade ≥3 infection rate normalised to the duration of the follow-up at day 500 (mean grade ≥3 infection rate 0.70 per year vs. 0.04 per year, *p* = 0.001). In addition, none of the patients in group 1 with lower aTreg frequencies and five patients in group 2, characterised by higher aTreg frequencies before the start of therapy, were hospitalised due to severe COVID-19 disease during follow-up (Figure 3B). 

No significant associations with the type of therapy (Table 2) or other clinical or laboratory parameters were seen (Table 1).

When aTreg percentages and CD4 counts were examined in sequential samples from patients on therapy, after initial decrease, the percentage of aTregs among FOXP3^+^CD4^+^ T cells from some patients from both groups even increased in the follow-up samples. In contrast, with the exception of one patient in each group, CD4 counts in the follow-up samples from patients in both groups were lower than before therapy (Figure 3C,D).

There was a significant negative correlation between CD4 counts and the aTreg frequencies in samples from treated patients during the follow-up (Figure 3E), but not in the samples from CLL patients with advanced disease at the start of therapy.

To examine the role of homeostatic cytokines in aTreg expansion, we investigated basal phosphorylation of STAT5 (pSTAT5) in CD4^+^ T cells in blood samples from some treated CLL patients during the follow-up. We found a significant correlation between the percentage of aTregs among FOXP3^+^CD4^+^ T cells and the pSTAT5 MFI in CD4^+^T cells (Figure 3E), suggesting that homeostatic STAT5 signalling could be responsible for the increased aTreg frequencies in the patients during therapy. 

### 3.4. Higher Basal STAT5 Phosphorylation Levels in CD4 T Cells from Patients with CLL Treated with Chemo-Immunotherapy

Basal CD4^+^ T cell pSTAT5 levels in whole-blood samples from CLL patients receiving therapy at the time of analysis were also compared to untreated CLL patients and HC. pSTAT5 MFI in CD4^+^T cells (Figure 4A), gated as shown on Figure 4B, was not significantly increased when all samples from treated CLL patients were compared to HC and patients not receiving any form of therapy. However, basal pSTAT5 levels were significantly higher when the samples from patients treated with chemoimmunotherapy (CIT) were compared to samples from patients receiving Bruton Tyrosine Kinase Inhibitor (BTKi) therapy (Figure 4C) and CLL patients not receiving therapy (*p* = 0.04) and HC (*p* = 0.02).

### 3.5. Relationship between STAT5 Phosphorylation and Ki-67 Expressing CD4 T Cell Subsets

To assess the relationship between the turnover homeostasis of aTregs and FOXP3^−^ Tcon CD4^+^ T cell subsets and STAT5 phosphorylation, the expression of Ki-67 was determined as an indicator of proliferation [38] in the peripheral blood of an aTreg subset and FOXP3^−^ Tcon. 

The percentage of Ki-67 expressing proliferating cells among CD4^+^ T cells of both an aTreg subset (Figure 4D) and a Tcon (Appendix A) were significantly increased in peripheral blood from CLL patients not receiving therapy and treated CLL patients compared to HC. Ki-67 expressing aTregs and Tcons were also significantly increased in treated patients compared to CLL patients not receiving therapy (Figure 4D).

Moreover, consistent with the CD4^+^ T cell pSTAT5 levels, the CLL patients treated with CIT also displayed higher levels of Ki-67 expressing aTregs (Figure 4E) and Tcons (Appendix A) compared to patients treated with BTKi.

To determine if basal STAT5 activation is involved in the turnover of Tcon and aTreg subsets from CLL patients treated with BTKi or CIT, we examined the relationship between the levels of Tcon / aTreg Ki-67 expressing subset and pSTAT5. Consistent with the crucial role of STAT5 signalling in the proliferation of Tcons, CD4^+^ T cell pSTAT5 MFI was significantly positively correlated with the percentage of Ki-67^+^ Tcons among CD4^+^ T cells in peripheral blood from treated patients with CLL (Appendix A). On the other hand, a significant correlation between proliferating aTreg subsets (percentage of Ki-67^+^ aTreg among CD4^+^ T cells) and CD4 T cell pSTAT5 MFIs was also found (Figure 4F). 

Together, our results suggested that augmented basal STAT5 signalling, found in patients treated with CIT, which was positively related to Ki-67 expression, may confer proliferative advantages to Tcons, and could also be involved in the increase in proliferating aTreg subsets. 

### 3.6. Differences in Basal STAT5 Phosphorylation between aTreg and Conventional T Cells

We then compared the basal levels of STAT5 phosphorylation in both aTreg and Tcon subsets from HC, therapy-naïve CLL patients and in BTKi- or CIT-treated patient samples. Compared with HC and untreated patients, pSTAT5 levels were higher but not significantly different from treated patients with CLL in both subsets. However, aTregs displayed significantly higher levels of pSTAT5 compared to Tcons in samples from treated but not untreated CLL patients or HC (Figure 4G). 

Therefore, due to the critical role of STAT5 signalling in Treg development, the significantly higher levels of basal pSTAT5 in the aTreg subset compared to FOXP3^−^ Tcon cells detected in treated patients with advanced CLL may be responsible for the perturbed homeostasis between the aTreg and Tcon subsets of CD4^+^ T cells.

While CIT patients had significantly higher pSTAT5 levels in both subsets compared to BTKi-treated CLL patients, Tcon pSTAT5 levels differed more significantly between the two groups of treated patients (Figure 4H). 

## 4. Discussion

Recent results obtained in the Em-TCL1 mouse model of CLL provide evidence that specific subsets of Tregs may be essential for leukemia progression in immunocompetent mice and can be efficiently targeted to block CLL progression. This Treg subset contributed indirectly to the proliferation of the CLL clone, by suppressing the proliferation of effector T cells which in turn suppress CLL cells [39].

Human CD4^+^FOXP3^+^ T cells contain cytokine-secreting nonsuppressive T cells that display a low expression of FOXP3, in the so called non-Treg fraction [15]. It is therefore critical to analyse the suppressive subpopulations, such as rTregs and aTregs, and not only total FOXP3^+^CD4^+^ T cells, when investigating their potential role in immune suppression.

Earlier studies, performed on peripheral blood samples from patients with CLL described elevated total Treg levels, which also correlated with worse prognostic factors and advanced disease stages [17,18,19,20].

Consistent with the results of recent studies, which also analysed FOXP3 expression [40,41], we found significant increases in the percentage of all FOXP3^+^ cells among CD4^+^ T cells from patients with CLL. 

However, the percentage of specific-activated Treg subsets of FOXP3^+^ cells, which were also characterised by the highest levels of CD25 expression, were significantly increased in the advanced Binet C stage of disease and were associated with a malignant B cell burden in our patients with CLL. 

Moreover, the aTreg subset, defined as either CD45RA^−^FOXP3^hi^ or CD45RA^−^CD25^hi^CD127^lo/−^ cells, was significantly positively correlated with the TTM-indicator of the tumour mass within all major body compartments in our patients. As aTreg cells are thought to be the main effectors of suppression among human FOXP3^+^ Treg subsets [15], their expansion in patients with advanced CLL could support tumour-immune evasion by suppressing antitumour T cell responses. The presence of cytotoxic T lymphocyte-associated protein 4 (CTLA-4) expression in aTreg cells, but not in rTreg cells [15], suggests that aTregs are involved in critical contact dependent suppression, as Treg cell-specific CTLA-4 deficiency impairs Treg cell suppressive function, not only in vitro, but also in vivo in mice [42]. In addition, increased CD4^+^CD25^+^FOXP3^+^CTLA-4^+^ Tregs occurring alongside leukemia development were described in the Eμ-TCL1 transgenic mouse model of CLL [43].

On the other hand, recent studies uncovered a key role for cytokine production by CLL cells in enhancing its Treg suppressive capacity by upregulating FOXP3 expression [44].

Changes in the cytokine environment and imbalances between effector CD4^+^ T cell subsets (e.g., Th17) and FOXP3^+^ Tregs have been the subject of research in CLL patients [45,46]. However, there were only a few attempts to define the altered Treg homeostasis based on the newer classifications of FOXP3^+^ Treg subpopulations using multicolour flow cytometry or mass cytometry in patients with CLL [47,48]. In addition, to our knowledge, there has been no report on homeostatic cytokine STAT5 signalling in Treg subsets in patients with CLL.

Activation of the STAT5 transcription factor downstream of the Interleukin-2 receptor (IL-2R) induces the expression of FOXP3, which is a critical step in the differentiation of Treg [49]; we also analysed STAT5 signalling/nuclear translocation by using multispectral imaging cytometry. We could show that after IL-2 stimulation, nuclear localisation of pSTAT5, measured using the Similarity score, was higher in both the FOXP3 high- and FOXP3 low-expressing cells compared to the FOXP3-negative cells.

Using conventional flow cytometry and anti-pSTAT5 antibodies, it has been shown before that FOXP3^+^ Tregs in whole blood from healthy donors differ from FOXP3^−^ Tcons in terms of their higher capacity to phosphorylate STAT5 proteins in response to IL-2 stimulation [25].

However, we found that STAT5 signalling responses after SARS-CoV-2 antigen-specific stimulation, measured at the pSTAT5 MFI fold change, were significantly different, even among Treg subsets. They were higher in aTregs than in rTregs, from both HC, as well as in patients with CLL. In addition, the frequency of aTreg, but not rTreg, subsets among CD4^+^FOXP3^+^ T cells was increased following such whole-blood SARS-CoV-2 antigen-specific stimulation. Together with the significant correlation between pSTAT5 and the frequency of aTreg, our findings indicate that aTreg-specific pSTAT5 responses, which are higher than in rTreg subsets, are also associated with the expansion of aTreg subsets among FOXP3^+^CD4^+^ T cells following whole-blood SARS-CoV-2 antigen-specific stimulation.

The early release of homeostatic cytokine IL-2 in response to antigens was followed by IL-2-dependent STAT5 phosphorylation, which is shown to occur primarily in FOXP3^+^ Tregs within hours of T cell priming in the mouse model in vivo. As the first responders to IL-2, Treg cells also multiplied and showed increased suppressive capacity [50].

Tregs were shown to influence the magnitude and severity of acute and chronic infections through the suppression of pathogen-specific Tcons and CD8^+^ cytotoxic T cells. Accordingly, the clearance of microbial pathogens was promoted by the depletion of Treg populations [51]. Of note, elevated Treg levels in CLL patients correlated with decreased T cell responses, not only against tumors, but also against microbial antigens in functional assays [52].

Consistent with this, the disease course of a subgroup of our CLL patients with higher aTreg percentages at the start of therapy was characterised by more frequent episodes of severe infections during follow-up. Five patients from the latter group also experienced severe COVID-19, which was shown in a recent study to entail a striking induction of FOXP3 expression in Tregs [35]. In the same study, CD45RA, which marks naïve-resting Tregs, was also reduced in patients with severe COVID-19. In addition, the so-called Severe COVID19 Treg Signature was associated with changes in several transcription factors previously associated with differential gene expression in activated Tregs, including the downregulation of BACH2 (BTB Domain, and CNC Homolog) [35,53].

Recent data demonstrate that BACH2 downregulation in normal lymphocytes increases age-related resistance to apoptosis and these alterations were even more pronounced in the T and B cells from CLL patients [54].

BACH2 binds to enhancers of genes involved in aTreg differentiation and represses their T cell receptor (TCR)-driven induction by competing with AP-1 (activating protein-1) factors for DNA binding [55]. Therefore, BACH2 could also be involved in the perturbed aTreg homeostasis found in our patients with CLL. In addition, the low expression of this transcription factor was recently shown to predict adverse outcomes in patients with CLL [56]. 

Appendix A shows the Kaplan–Meier curves for the overall and progression-free survival of the two groups of our patients in the follow-up study. However, our study has limitations in this regard, as conclusions on the survival or relapse development, as classified according to the Tregs subsets analysis, would require at least a longer follow-up with the patients.

In our patients, higher aTreg percentages among FOXP3^+^CD4^+^ T cells at the start of therapy were associated with a disease course characterised by more frequent serious infections during the follow-up with our patients with CLL.

However, the significant correlations found in our treated patients between the follow-up pSTAT5 levels and aTreg, but not the rTreg frequency, mirror results of in vitro SARS-CoV2 Ag-specific activation, where an increase in pSTAT5 levels was associated with the expansion of aTregs, but not rTreg subsets, despite the latter also showing increased pSTAT5 levels.

Although we did not use pharmacological inhibitors of STAT5 signalling in our experiments, we did find that after anti-CD3/CD28 stimulation in vitro, both the pSTAT5 levels (MFI) and the FOXP3 expression (MFI) in the CD25^+^FOXP3^+^ subset of magnetically sorted CD4^+^ T cells were IL-2-dependent, as they were both significantly decreased with neutralising anti-IL-2 antibodies (Appendix A).

Therefore, STAT5 signalling, implicated in both FOXP3 expressions [49], as well as the maintenance of the Treg suppressive function (24), may be even more important in the homeostasis of the most suppressive aTreg subsets of CD4^+^ T cells, characterised by the highest levels of FOXP3 expression. 

Although the CD4 counts in the follow-up samples from most of our patients were lower than before therapy, the percentage of aTreg among FOXP3^+^CD4^+^ T cells increased in the follow-up samples, even in some of the patients who were already characterised by high aTreg frequencies at the start of follow-up. In addition, lower CD4 T cell counts in follow-up samples from CLL patients were associated (significantly negatively correlated) with higher follow-up aTreg frequencies. As there was also a significant correlation between the percentage of aTreg among FOXP3^+^CD4^+^ T cells and the pSTAT5 levels in CD4^+^ T cells from individual patients during the follow-up, homeostatic STAT5-signalling, associated with lower CD4 counts, could be responsible for the increased frequencies of aTregs during therapy.

The reduced CD4 counts in patients with CLL after frontline fludarabine, cyclophosphamide and rituximab (FCR) therapy were not associated with a risk of infections. In this study, according to the authors’ suggestion, the CD4^+^ T cells mostly consisted of regulatory Tregs [57]. The impact of CIT on total Tregs was shown in another recent study, where the relative number of Tregs remained higher than in the controls because other CD4^+^ T cells numbers decreased more significantly [58]. An increased regulatory T cell to CD4 ratio after FCR was observed. Although a higher percentage of total Treg among the CD4^+^ T cells after 6 months of FCR therapy correlated negatively with infections, the relative contribution of individual Treg subsets was less clear, as activated Tregs were defined only by the expression of HLADR [59]. 

In our study, increased CD4^+^ T cell STAT5 phosphorylation was found only in patients on CIT, but not untreated patients, compared to HC. Basal pSTAT5 levels were also significantly higher in patients treated with CIT than in patients on BTKi therapy, suggesting that homeostatic cytokine STAT5 signalling is more increased in the former patient group.

Consistent with that, CLL patients treated with CIT also displayed significantly higher levels of Ki-67 expressing Tcon and aTreg subsets compared to patients treated with BTKi. Further supporting the role of STAT5 activation in the increased turnover of Tcons and aTregs from CLL patients on therapy, the pSTAT5 levels in CD4^+^ T cells were also significantly positively correlated with the percentage of Ki-67^+^ Tcons, as well as with the percentage of Ki-67^+^ aTregs among CD4^+^ T cells. Therefore, our results suggested that while STAT5 signalling, which was more significantly correlated to Tcon Ki-67 expression, may confer a proliferative advantage to conventional CD4^+^ T cells, STAT5 activation/phosphorylation could also be involved in increasing the proliferation of aTreg subset. 

Although basal STAT5 signalling was significantly increased in both subsets of CD4^+^ T cells, when patients treated with CIT were compared to patients on BTKi therapy, the basal STAT5 phosphorylation levels were also significantly higher in aTregs compared to Tcons from treated patients with CLL. Such imbalanced homeostatic cytokine signalling could also confer survival advantages to aTregs over Tcons during therapy, as the activation of a STAT5 signalling pathway up-regulates anti-apoptotic protein Bcl-2 [60]. Indeed, T cells with a tuned-up survival state, as indicated by a high Bcl-2 content enriched with Treg subsets, were described recently in patients with CLL [61].

Recent studies reveal a statistically higher percentage of Treg cells in the bone marrow than in the peripheral blood in the group of children with acute lymphoblastic leukemia [62]. Of note, among Treg subpopulations, only memory Tregs showed statistically significant differences (REF). In addition, in another study, the reconstituted bone marrow-residing CD4^+^CD25^+^FOXP3^+^ Treg of the myeloma patients after allogeneic stem cell transplantation consisted preferably of CD45RA^−^CCR7- memory T cells. They also expressed high levels of cytotoxic T-lymphocyte antigen 4 and showed a strong inhibitory function [63]. Together, the results of latter studies suggest that most phenotypical suppressive subsets can be expanded, not only in peripheral blood, but also in the bone marrow compartment in patients with other haematological malignancies in addition to CLL [62,63,64].

## 5. Conclusions

Together, our findings indicate that specific-activated Treg subsets are expanded in patients with advanced disease and increase in the frequency of this subset among CD4^+^ T cells is associated with the TTM indicator of the tumour mass within all major body compartments in patients before therapy. An increased aTreg fraction could represent a possible marker of more severe CLL disease, characterised by serious infectious complications, as the disease course of a subgroup of our CLL patients with higher aTreg percentages among CD4^+^FOXP3^+^ T cells at the start of therapy was characterised by more frequent episodes of severe infections during the follow-up. Augmented homeostatic STAT5 signalling could support aTreg expansion, as higher pSTAT5 levels were significantly correlated with increased aTreg percentages among CD4^+^FOXP3^+^ T cells following SARS-CoV-2 antigen-specific stimulation in vitro, as well as during the follow-up of the patients on therapy. While the basal phosphorylation of STAT5 was significantly increased in CD4^+^ T cells from patients on CIT and was positively correlated with both proliferating Tcon and aTreg subsets, basal pSTAT5 levels were also significantly higher in aTregs compared to Tcons from treated patients with CLL. Therefore, dysbalanced STAT5 signalling may be involved in perturbed Tcon/aTreg homeostasis in CLL patients on therapy.

## Figures and Tables

**Figure 1 cancers-16-03228-f001:**
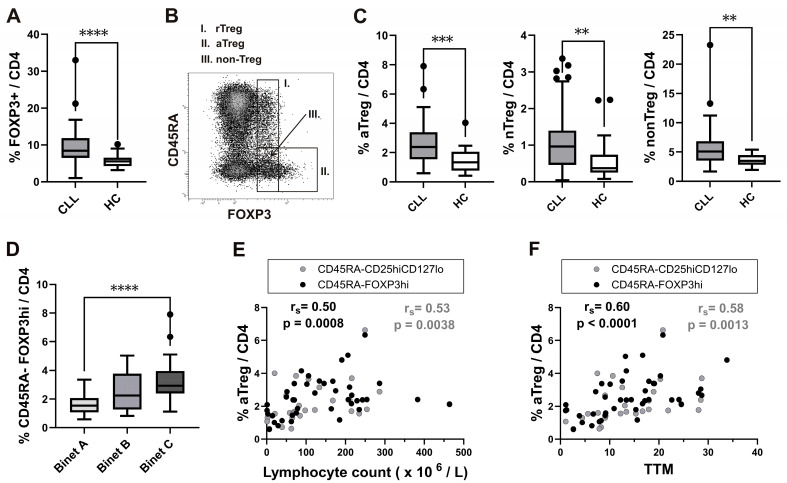
Increased frequency of the activated Treg subset in peripheral blood from CLL patients with untreated advanced disease and correlation with total tumour mass (TTM) scoring. (**A**) Box-and-whisker plot shows the percentage of FOXP3^+^ cells among gated CD4^+^ T cells from the patients with CLL (CLL, *n* = 54) and healthy controls (HC, *n* = 20). (**B**) FOXP3^+^ cells among gated CD4^+^ T cells were subdivided into three fractions based on CD45RA and the level of FOXP3 expression: (I) CD45RA^+^ FOXP3^lo^ rTreg, (II) CD45RA^−^ FOXP3^hi^ aTreg, and (III) CD45RA^−^ FOXP3^lo^ non-Treg subsets, as shown on the representative dot plot. (**C**) Percentage of aTreg, non-Treg, and rTreg among CD4^+^ T cells from patients with CLL (*n* = 54) and HCs (*n* = 20). (**D**) Percentage of aTregs defined as CD45RA^−^ FOXP3^hi^ among the CD4^+^ T cells from patients with CLL in the Binet stage A (*n* = 17), patients with CLL in the Binet stage B (*n* = 12) and patients with CLL in the Binet stage C (*n* = 25). (**E**) Correlation between the percentage of aTreg defined as either CD45RA^−^ FOXP3^hi^ (black symbols, *n* = 42) or CD45RA^−^ CD25^hi^ CD127^lo^/^−^ (grey symbols, *n* = 28) and the lymphocyte counts (×10^6^/L) from patients with CLL. (**F**) Correlation between the percentage of aTregs defined as either CD45RA ^−^ FOXP3^hi^ (black symbols, *n* = 42) or CD45RA^−^ CD25^hi^ CD127^lo/−^ (grey symbols, *n* = 28) and the TTM score in patients with CLL. r_s_, Spearman correlation coefficient; **** *p* < 0.0001.; *** *p* < 0.001; ** *p* < 0.01.

**Figure 2 cancers-16-03228-f002:**
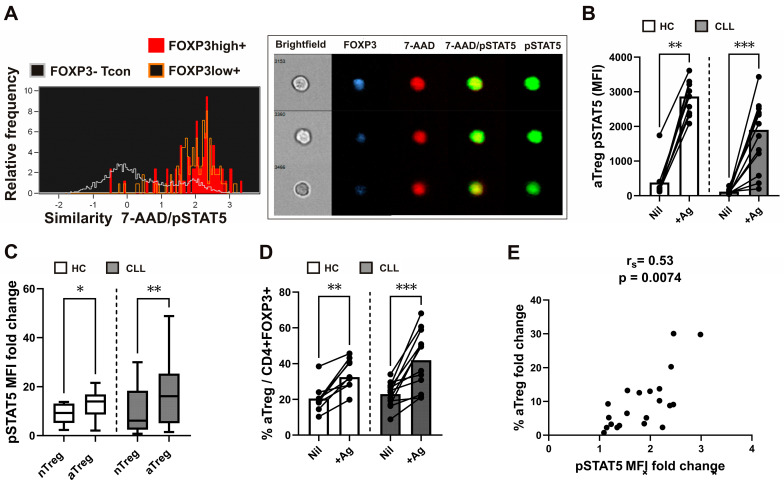
IL-2-induced pSTAT5/nuclear translocation in FOXP3^+^ cells, augmenting aTreg STAT5 signalling responses and correlation with increased proportions of aTregs following SARS-CoV-2 antigen-specific stimulation. (**A**) Representative histograms of 7-AAD/pSTAT5 Similarity scores in FOXP3 high (red) and FOXP3 low expressing (orange) cells as compared to FOXP3^−^ cells (white). Sample cell pictures are displayed on the right. In FOXP3-expressing (blue) cells, pSTAT5 (green) is preferentially localised to the nucleus stained with 7-AAD (red). Cell brightfield pictures are also displayed. (**B**) Whole-blood samples were obtained and stimulated or incubated for 16–24 h in tubes from the Quantiferon SARS-CoV-2 kit. Blood aliquots were taken from the Ag tubes (+Ag) and the negative control (Nil) tubes, aTreg pSTAT5 levels (MFI) are shown on plot. Patients with CLL (*n* = 12) are contrasted with healthy controls (*n* = 9). The data are presented as mean + SD. (**C**) STAT5 signalling responses in rTreg and aTreg are compared using the pSTAT5 MFI fold change, which is calculated by dividing the pSTAT5 MFI in the stimulated tube by the pSTAT5 MFI in the control tube from healthy controls (*n* = 9) and patients with CLL (*n* = 12). (**D**) Cumulative aTreg percentages among FOXP3^+^CD4^+^ T cells in whole-blood aliquots withdrawn from the negative control (Nil) and from the Ag tubes (+Ag). Healthy controls (*n* = 9) are compared to patients with CLL (*n* = 12). Data are expressed as mean with SD. (**E**) Correlation between the aTreg STAT5 signalling responses, measured as the pSTAT5 MFI fold change (pSTAT5 MFI in stimulated tube divided by pSTAT5 MFI in control tube) and fold change in the percentage of aTreg (percentage of aTreg among FOXP3^+^CD4^+^ T cells in stimulated tube divided by percentage of aTreg among FOXP3^+^CD4^+^ T cells in control tube) from healthy controls (*n* = 9) and patients with CLL (*n* = 12). r_s_, Spearman correlation coefficient; *** *p* < 0.001; ** *p* < 0.01; * *p* < 0.05.

**Figure 3 cancers-16-03228-f003:**
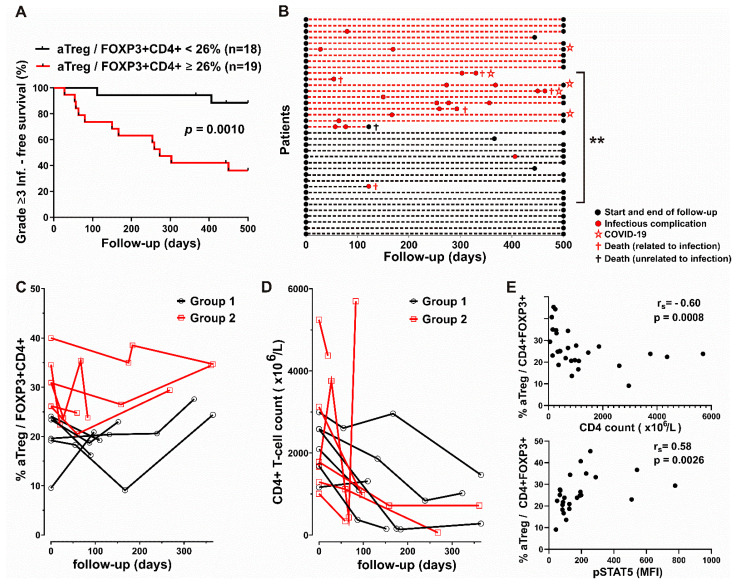
Significantly different disease courses in subgroups of patients defined based on aTreg frequencies and correlation between aTreg, CD4 counts and STAT5 signalling during follow-up. (**A**) Survival curve (Kaplan–Meier plot) showing shorter time to first grade ≥ 3 infection in group 2 patients with CLL with higher aTreg percentages (≥26%) among CD4^+^FOXP3^+^ T cells at start of follow-up. (**B**) Disease courses of 37 patients with CLL (*y* axis). Colour of dotted lines reflects subgroup designations. Increased flare frequency in group 2 (red dashed lines) when followed up to 500 d. (**C**,**D**) aTreg frequency (percentage of aTreg among FOXP3^+^CD4^+^ T cells) and CD4 counts in group 1 of patients with CLL (black symbols) vs. group 2 of patients with CLL (red symbols). (**E**) Correlation between percentage of aTreg among FOXP3^+^CD4^+^ T cells in samples from patients with CLL and CD4 counts during follow-up (upper panel, *n* = 28). Correlation between percentage of aTreg among CD4^+^ FOXP3^+^ T cells in samples from patients with CLL and basal pSTAT5 levels (MFI) in CD4^+^ T cells during follow-up (lower panel, *n* = 25). r_s_, Spearman correlation coefficient; ** *p* = 0.001–0.01.

**Figure 4 cancers-16-03228-f004:**
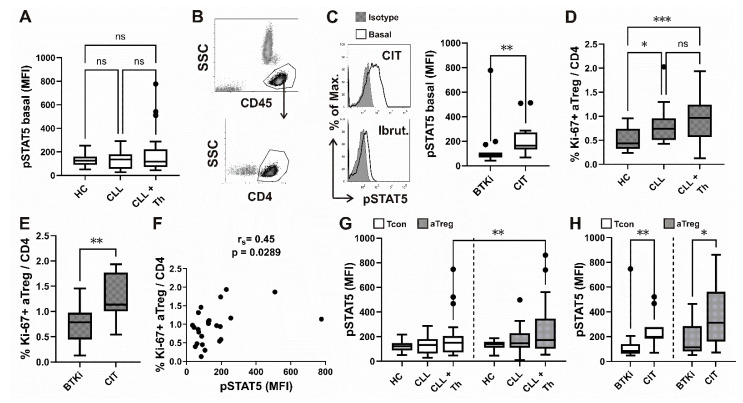
Higher basal pSTAT5 levels in CD4 T cells from patients with CLL treated with CIT, correlation with Ki-67+ aTreg and subset specific differences in basal STAT5 phosphorylation. (**A**) Box-and-whisker plot shows basal STAT5 phosphorylation (pSTAT5) levels (MFI) in CD4^+^ T cells from samples of untreated patients with CLL (CLL, *n* = 23), patients with CLL on CIT or therapy with BTKi (CLL + Th, *n* = 28) and HCs (*n* = 18). (**B**) Lymphocytes were gated as shown on CD45 vs. SSC density plot. CD4^+^ T cells were identified by first gating on lymphocytes and then on CD4^+^ cells. (**C**) pSTAT5 levels (MFI) in CD4^+^ T cells from samples of patients with CLL on CIT (CIT, *n* = 13) and therapy with BTKi (BTKi, *n* = 15). Representative histograms of pSTAT5 in gated CD4^+^ T cells from sample of patient with CLL on CIT and patient with CLL on therapy with BTKi-Ibrutinib are shown left: basal, untreated cells are compared with cells stained with isotype control. (**D**) Percentage of Ki-67^+^ aTreg cells (%Ki-67^+^ aTreg) among CD4^+^ T cells from samples of untreated patients with CLL (CLL, *n* = 17), patients with CLL on CIT or therapy with BTKi (CLL+Th, *n* = 25) and HCs (*n* = 17). (**E**) Percentage of Ki-67^+^ aTreg cells among CD4^+^ T cells from samples of patients with CLL on CIT (*n* = 10) or therapy with BTKi (*n* = 15). (**F**) Correlation between percentage of Ki-67^+^ aTreg cells and basal pSTAT5 levels (MFI) in CD4^+^ T cells from samples of patients with CLL on CIT or therapy with BTKi (*n* = 24). (**G**) Basal pSTAT5 levels (MFI) in aTreg subset (shown in gray) gated as shown on Figure 1B and FOXP3 negative Tcons (white) in samples from untreated patients with CLL (CLL, *n* = 21), patients with CLL on CIT or therapy with BTKi (CLL + Th, *n* = 24) and HCs (*n* = 17). (**H**) Basal pSTAT5 levels (MFI) in aTreg subset and FOXP3 negative Tcons in samples from patients with CLL on CIT (*n* = 11) or therapy with BTKi (*n* = 13). r_s_, Spearman correlation coefficient; ns, not significant; *** *p* < 0.001; ** *p* < 0.01; * *p* < 0.05.

**Table 1 cancers-16-03228-t001:** Demographic, clinical and laboratory data at the time of the follow-up study entry.

Parameter	Group 1 ^a^	Group 2 ^a^	*p*	Adjusted *p*
Cohort size	18	19	NA	NA
Age (y)	68 (3)	70 (2)	0.93	NS
Gender	5 F/13 M	8 F/11 M	0.49	NS
Ethnicity	19 Slovene	19 Slovene	NA	NA
Binet stage C	10/18	15/19	0.17	NA
Disease duration (mo)	49 (15)	60 (11)	0.32	NS
Age at diagnosis (y)	64 (3)	65 (2)	0.89	NS
TTM score t0	16.4 (1.8)	17.0 (1.7)	0.81	NS
TD score t0	0.77 (0.05)	0.75 (0.04)	0.69	NS
LN t0 (cm)	1.8 (0.4)	3.1 (0.8)	0.25	NS
Spleen t0 (cm)	1.7 (0.9)	2.2 (1.1)	0.60	NS
Lymphocytes t0 (×10^9^/L)	170.8 (30.5)	146.5 (19.2)	0.68	NS
Neutrophils t0 (×10^9^/L)	3.7 (0.6)	3.7 (0.4)	0.89	NS
CD4 count t0 (×10^3^/L)	2129 (238)	2445 (300)	0.77	NS
CD4% t0 (%)	3.7 (1.6)	2.1 (0.3)	0.48	NS
TP53 mutation	4/18	1/19	0.18	NS
Unmutated IGHV	11/18	8/19	0.33	NS
AIHA	2/18	2/19	>0.99	NS
Preexisting CLL therapy	7/18	5/19	0.49	NS
Hgb t0 (g/L)	104 (4)	108 (7)	0.96	NS
Tr t0 (×10^9^/L)	158 (23)	128 (12)	0.43	NS

^a^ *p* value refers to the comparison of group 1 (aTreg < 26% among CD4^+^FOXP3^+^ T cells) vs. group 2 (aTreg ≥ 26% CD4^+^FOXP3^+^ T cells); adjusted *p*—Bonferroni-adjusted *p* value; LN t0—diameter of largest palpable lymph node at time of enrolment; Spleen t0—palpable spleen below left costal margin node at time of enrolment; CD4%—percentage of CD4^+^ T cells among lymphocytes; AIHA—autoimmune haemolytic anaemia; F—female; M—male; NA—not applicable; NS, *p* > 0.05; Preexisting CLL therapy—any CIT or BTKI therapy of CLL before enrolment; Hgb t0—Haemoglobin concentration at time of enrolment; t0—at time zero enrolment.

**Table 2 cancers-16-03228-t002:** Therapy during follow-up.

Therapy	Combinations	Group 1 ^a^		Group 2 ^a^		*p*	Adjusted *p*
		*n*/*N*	%	*n*/*N*	%		
CIT	All	4/18	25	8/19	42	0.29	NS
	FCR	0/18	0	4/19	21	0.10	NS
	Chlorambucil + Rituximab	2/18	11	4/19	21	0.66	NS
	Chlorambucil + Obinutuzumab	1/18	5	0/19	0	0.49	NS
	Bendamustine + Rituximab	1/18	5	0/19	0	0.49	NS
BTKi	All	11/18	61	8/19	42	0.33	NS
	Ibrutinib	5/18	28	6/19	32	>0.99	NS
	Acalabrutinib	4/18	22	2/19	10	0.40	NS
	Acalabrutinib + Obinutuzumab	2/18	11	0/19	0	0.23	NS
Venetoclax	All combinations	3/18	17	3/19	16	>0.99	NS
	+Rituximab	0/18	0	2/19	10	0.49	NS
	+Obinutuzumab	1/18	5	0/19	0	>0.99	NS
	+Bendamustine + Obinutuzumab	2/18	11	1/19	5	0.60	NS

^a^ No. of patients/whole group; *p* value refers to the comparison of group 1 (aTreg ≤ 26% of CD4^+^FOXP3^+^ T cells) vs. group 2 (aTreg > 26% of CD4^+^FOXP3^+^ T cells); adjusted *p*—Bonferroni adjusted *p* value; therapy given at the start of and during follow-up; FCR—Fludarabine + Ciclophosphamide + Rituximab combination; NS, *p* > 0.05.

## Data Availability

Datasets used in this article are available from the corresponding author on reasonable request.

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
