# Peer review of "Increased Frequency of Circulating Activated FOXP3+ Regulatory T Cell Subset in Patients with Chronic Lymphocytic Leukemia Is Associated with the Estimate of the Size of the Tumor Mass, STAT5 Signaling and Disease Course during Follow-Up of Patients on Therapy"

_cancers, 2024, doi:10.3390/cancers16183228_

Round 1
Reviewer 1 Report
Comments and Suggestions for Authors
What is the impact of pharmacological inhibition of STAT5 in Tregs? A validating experiment would be very helpful to the reader.
Comments on the Quality of English LanguageGood.
Author Response
Reviewer 1 Comments and Suggestions for Authors
What is the impact of pharmacological inhibition of STAT5 in Tregs? A validating experiment would be very helpful to the reader.
Response:
We would like to thank reviewer for pointing to the important question on the impact of pharmacological inhibition of STAT5 in Tregs. Although we did not use pharmacological inhibitors of STAT5 signaling in our experiments, we did find that after anti-CD3/CD28 stimulation in vitro, both pSTAT5 levels (MFI) and FOXP3 expression (MFI) in CD25+FOXP3+ subset of magnetically sorted CD4+ T-cells were IL-2-dependent, as they were both significantly decreased with neutralizing anti-IL-2 antibodies. In addition, significant correlation between the increased aTreg pSTAT5 MFI and increased percentage of this subset of Tregs was found following antigen-specific stimulation, suggesting that cytokine dependent STAT5 signaling contributed to aTreg expansion in vitro. The correlation is shown in the Figure 2E in the original manuscript and the new data after anti-CD3/CD28 stimulation in vitro are included the new Supplementary figure S6.
Please note that the methods used for inhibition of pSTAT5 with neutralising anti-IL2 antibodies after anti-CD3/CD28 stimulation were described in our previous publication (reference: Roškar Z, Dreisinger M, Tič P, Homšak E, Bevc S, Goropevšek A. New Flow Cytometric Methods for Monitoring STAT5 Signaling Reveal Responses to SARS-CoV-2 Antigen-Specific Stimulation in FOXP3+ Regulatory T Cells also in Patients with Advanced Chronic Lymphocytic Leukemia. Biosensors (Basel). 2023, 13(5), 539.) and are now included in the Supplementary materials together with methods describing.
Reviewer 2 Report
Comments and Suggestions for Authors
This is an interesting and extensive study about the increased frequency of circulating activated FOXP3+ T cell subset in CLL patients in relation to the size of tumor mass, STAT5 signaling and disease course during the follow up of patients with or without treatment.
There are minor typing errors, spelling errors, repeated or missing words. The authors should check the manuscript very carefully.
As there are a lot of results perhaps a table -summary of the most important results in the different groups of patients might provide a clear picture.
The authors don't explain why they decided to use the percentage of 26% of activated Tregs among CD4+FOXP3+ cells to form the two groups of patients. How did this percentage arise as an important differentiating marker?
In the methods all monoclonal antibodies are presented as conjugated with many different fluorochromes. The authors should explain the change of the fluorochromes or perhaps present the panels (or protocols) used.
In 2.4 CD4+ epitopes from the spike protein or CD4+CD8+ epitopes from the spike protein. The epitopes are not on CD4+ or CD8+ cells but on the spike protein. The authors should explain the meaning or rephrase it.
The legends of the figures are very extensive. Perhaps the authors could remove some of the methods information included in the legends e.g. fig 2.
Could the percentage of aTregs be used as a prognostic marker?
Comments on the Quality of English Language
The English is quite good but there are minor typing mistakes. Some words are missing and some are repeated.
Examples:
line233 to account to account
line 301: in not fin
line 517: in in
Author Response
Reviewer 2 Comments and Suggestions for Authors
This is an interesting and extensive study about the increased frequency of circulating activated FOXP3+ T cell subset in CLL patients in relation to the size of tumor mass, STAT5 signaling and disease course during the follow up of patients with or without treatment.
There are minor typing errors, spelling errors, repeated or missing words. The authors should check the manuscript very carefully.
As there are a lot of results perhaps a table -summary of the most important results in the different groups of patients might provide a clear picture.
Response: We tried our best to summarize our work in a graphic summary.
The authors don't explain why they decided to use the percentage of 26% of activated Tregs among CD4+FOXP3+ cells to form the two groups of patients. How did this percentage arise as an important differentiating marker?
Response: The cutoff value for the percentage of activated Tregs was determined as the means + SEM (26,3%) in HCs. We have now added this explanation in text on page #... of the revised manuscript.
In the methods all monoclonal antibodies are presented as conjugated with many different fluorochromes. The authors should explain the change of the fluorochromes or perhaps present the panels (or protocols) used.
Response: As suggested by the reviewer, typical panel used and analysed on the FACSymphony A3 flow cytometer for determination of pSTAT5 levels (MFI) in aTreg was included in the new supplementary figure S1.
In 2.4 CD4+ epitopes from the spike protein or CD4+CD8+ epitopes from the spike protein. The epitopes are not on CD4+ or CD8+ cells but on the spike protein. The authors should explain the meaning or rephrase it.
Response: We have rephrased this paragraph in the original in the original manuscript to now read: “Epitopes from the spike protein's S1 subunit for CD4+ T-cells are present in the SARS-CoV-2 Ag1 tube, and epitopes from the spike protein's S1 and S2 subunits for CD4+ and CD8+ T-cells are present in the SARS-CoV-2 Ag2 tube.2. “ in the revised manuscript.
The legends of the figures are very extensive. Perhaps the authors could remove some of the methods information included in the legends e.g. fig 2.
Response: We remove some text as suggested.
Could the percentage of aTregs be used as a prognostic marker?
Response: The percentage aTregs must be validated in further studies before it can be used as a prognostic marker.
The English is quite good but there are minor typing mistakes. Some words are missing and some are repeated.
Response: We carefully checked the manuscript.
Examples:
line233 to account to account
line 301: in not fin
line 517: in in
Reviewer 3 Report
Comments and Suggestions for Authors
The study by Zlatko Roskar and co-authors provides an insightful analysis of Treg subsets in CLL patients. The scientific background is highly reasonable as Tregs play an essential role in anti-cancer immunity, and attempts to break down their cancer-protective role seem to be a promising challenge in anti-cancer therapy. The paper is well-structured and supplemented with high-quality figures and supplementary data. The introduction gives a solid background for presented data, and detailed methodology, especially flow cytometry, enables repeatability of experiments. Another solid advantage is the Treg subsets classification by CD45RA and FoxP3 expression, which is up-to-date and in line with current trends in immunology. What's more, Treg functional testing and phosphorylation assessment were done well. There are only some minor flaws that limit results and conclusions.
1. The most critical limitation is the lack of overall patient survival or relapse development as classified according to the Tregs subsets analysis.
2. Please clarify the study group information, as four different populations have been pointed out in the text, "Group 1", "Group 2", healthy control and laboratory staff members, but detailed information was only given to Groups 1 and 2 in Table 1. For patients and healthy control demographics and clinical data would be essential. Please add supplementary data on patients' and controls' inclusion/exclusion criteria. Were there any CLL patients with a history of autoimmunity?
3. Please add a representative example of a flow cytometry gating strategy for control and patient samples, especially for MFI testing. How was 15 min cells' stimulation with IL-2 selected? Please refer to the published protocol or authors' validation data.
4. Some parts of the result section were suitable for study design or discussion. See examples like page 8, lines 294 – 299, or page 9, lines 358 – 362. Please reorganize the results section.
5. In Figure 3A, please specify that this was a COVID-19 infection-free survival. Figures 3C and 3D are challenging to follow; please consider changing the graph type to be more affordable. Figure 4B antigen identification is missing on the cytogram.
6. Please add a short paragraph in the discussion section on Tregs infiltrating the bone marrow compartment in leukaemias to specify some differences/similarities with your data obtained from peripheral blood. Please cite at least the following:
· J Immunol Res. 2018; 2018: 1292404 ; PMCID: PMC5996432 ; PMID: 30003111 ; DOI: 10.1155/2018/1292404
· Leuk Lymphoma 2018 Feb;59(2):486-489. doi: 10.1080/10428194.2017.1330475. Epub 2017 Jun 2. ; PMID: 28573905 ; DOI:10.1080/10428194.2017.1330475
· Haematologica. 2008 Mar;93(3):423-30. doi: 10.3324/haematol.11897. Epub 2008 Feb 20. ; PMID: 18287134 ; DOI:10.3324/haematol.11897
Author Response
Reviewer 3 Comments and Suggestions for Authors
The study by Zlatko Roskar and co-authors provides an insightful analysis of Treg subsets in CLL patients. The scientific background is highly reasonable as Tregs play an essential role in anti-cancer immunity, and attempts to break down their cancer-protective role seem to be a promising challenge in anti-cancer therapy. The paper is well-structured and supplemented with high-quality figures and supplementary data. The introduction gives a solid background for presented data, and detailed methodology, especially flow cytometry, enables repeatability of experiments. Another solid advantage is the Treg subsets classification by CD45RA and FoxP3 expression, which is up-to-date and in line with current trends in immunology. What's more, Treg functional testing and phosphorylation assessment were done well. There are only some minor flaws that limit results and conclusions.
- The most critical limitation is the lack of overall patient survival or relapse development as classified according to the Tregs subsets analysis.
Response: We have included new Supplementary figure S5 with Kaplan-Meier curves for overall and progression free survival of the two groups of our patients in the follow-up study.
We have also included a paragraph in the discussion (page 15, lines 762), noting the limitation of our study, as conclusions on survival or relapse development as classified according to the Tregs subsets analysis would require at least longer follow-up of patients.
- Please clarify the study group information, as four different populations have been pointed out in the text, "Group 1", "Group 2", healthy control and laboratory staff members, but detailed information was only given to Groups 1 and 2 in Table 1. For patients and healthy control demographics and clinical data would be essential. Please add supplementary data on patients' and controls' inclusion/exclusion criteria. Were there any CLL patients with a history of autoimmunity?
Response: We have included Supplementary data on controls' demographics and exclusion criteria (no history of allergies, acute infections, autoimmune disorders, or immunosuppressive medicines) in the new Supplementary table 2. There were no patients with a history of autoimmunity before their diagnosis of CLL. The data on autoimmune manifestation (AIHA) are shown in the Tables 1 and 2 of the original manuscript. We have included the data on AIHA also in the revised Supplementary table 1, which describes patients not included in the follow-up (Tables 1 and 2).
- Please add a representative example of a flow cytometry gating strategy for control and patient samples, especially for MFI testing. How was 15 min cells' stimulation with IL-2 selected? Please refer to the published protocol or authors' validation data.
Response: As suggested by the reviewer, gating hierarchy for determination of pSTAT5 levels (MFI) in aTreg was included in the new supplementary figure S1.
Maximal pStat5 induction by T cells was observed before in different published Phosflow protocols for stimulation with IL-2 at 15 min. As suggested by the reviewer, we have included example of such reference: Castro I, Yu A, Dee MJ, Malek TR. The basis of distinctive IL-2- and IL-15-dependent signaling: weak CD122-dependent signaling favours CD8+ T central-memory cell survival but not T effector-memory cell development. J Immunol. 2011 Nov 15;187(10):5170-82.
- Some parts of the result section were suitable for study design or discussion. See examples like page 8, lines 294 – 299, or page 9, lines 358 – 362. Please reorganize the results section.
Response: We have upgraded the results.
- In Figure 3A, please specify that this was a COVID-19 infection-free survival. Figures 3C and 3D are challenging to follow; please consider changing the graph type to be more affordable. Figure 4B antigen identification is missing on the cytogram.
Response: In Figure 3A, we have specified that Kaplan-Meier curve depicts Grade ≥ 3 infection-free survival. Please note that patients who experienced severe COVID-19 infection during follow-up, are shown with symbols on Figure 3B. We have increased the size of symbols and fonts in Figures 3C and 3D to make them less challenging to follow. Figure 4B antigen identification was also added on the cytogram.
- Please add a short paragraph in the discussion section on Tregs infiltrating the bone marrow compartment in leukaemias to specify some differences/similarities with your data obtained from peripheral blood. Please cite at least the following:
- J Immunol Res. 2018; 2018: 1292404 ; PMCID: PMC5996432 ; PMID: 30003111 ; DOI: 10.1155/2018/1292404
- Leuk Lymphoma 2018 Feb;59(2):486-489. doi: 10.1080/10428194.2017.1330475. Epub 2017 Jun 2. ; PMID: 28573905 ; DOI:10.1080/10428194.2017.1330475
- Haematologica. 2008 Mar;93(3):423-30. doi: 10.3324/haematol.11897. Epub 2008 Feb 20. ; PMID: 18287134 ; DOI:10.3324/haematol.11897
Response: As suggested by the reviewer, a short paragraph was added on page #... in the discussion section on Tregs infiltrating the bone marrow compartment in leukaemias to specify some differences/similarities with our data obtained from peripheral blood.
We have also included the suggested references:
Niedźwiecki M, Budziło O, Zieliński M, Adamkiewicz-Drożyńska E, Maciejka-Kembłowska L, Szczepański T, Trzonkowski P. CD4+CD25highCD127low/-FoxP3+ Regulatory T Cell Subpopulations in the Bone Marrow and Peripheral Blood of Children with ALL: Brief Report. J Immunol Res. 2018 May 29;2018:1292404.
Lad D, Hoeppli R, Huang Q, Garcia R, Xu L, Toze C, Broady R, Levings M. Regulatory T-cells drive immune dysfunction in CLL. Leuk Lymphoma. 2018 Feb;59(2):486-489.
Atanackovic D, Cao Y, Luetkens T, Panse J, Faltz C, Arfsten J, Bartels K, Wolschke C, Eiermann T, Zander AR, Fehse B, Bokemeyer C, Kroger N. CD4+CD25+FOXP3+ T regulatory cells reconstitute and accumulate in the bone marrow of patients with multiple myeloma following allogeneic stem cell transplantation. Haematologica. 2008 Mar;93(3):423-30.